# A Polyphenol Enriched Variety of Apple Alters Circulating Immune Cell Gene Expression and Faecal Microbiota Composition in Healthy Adults: A Randomized Controlled Trial

**DOI:** 10.3390/nu13041092

**Published:** 2021-03-27

**Authors:** Matthew P. G. Barnett, Wayne Young, Kelly Armstrong, Diane Brewster, Janine M. Cooney, Stephanie Ellett, Richard V. Espley, William Laing, Paul Maclean, Tony McGhie, Greg Pringle, Nicole C. Roy, Lynnette R. Ferguson

**Affiliations:** 1AgResearch Limited, Grasslands Research Centre, Palmerston North 4442, New Zealand; wayne.young@agresearch.co.nz (W.Y.); kelly.armstrong@esr.cri.nz (K.A.); paul.maclean@agresearch.co.nz (P.M.); nicole.roy@otago.ac.nz (N.C.R.); 2Riddet Institute, Palmerston North 4442, New Zealand; 3The High-Value Nutrition National Science Challenge, Auckland 1023, New Zealand; 4The New Zealand Institute for Plant and Food Research, Auckland 1025, New Zealand; di.brewster@plantandfood.co.nz (D.B.); richard.espley@plantandfood.co.nz (R.V.E.); greg.pringle@plantandfood.co.nz (G.P.); 5The New Zealand Institute for Plant and Food Research, Hamilton 3214, New Zealand; janine.cooney@plantandfood.co.nz; 6Discipline of Nutrition and Dietetics, The University of Auckland, Private Bag 92019, Auckland 1142, New Zealand; steph@ellett.co.nz; 7The New Zealand Institute for Plant and Food Research, Palmerston North 4410, New Zealand; william.laing@plantandfood.co.nz (W.L.); tony.mcghie@plantandfood.co.nz (T.M.); 8Liggins Institute, The University of Auckland, Auckland 1023, New Zealand; 9Department of Human Nutrition, University of Otago, Dunedin 9016, New Zealand

**Keywords:** *Malus* × *domestica*, RNA-Seq, 16s rRNA, immunoglobulins

## Abstract

Polyphenols within fruits and vegetables may contribute to health benefits due to their consumption, with the anthocyanin sub-set also adding colour. The Lemonade™ apple variety has green skin and white flesh, with low anthocyanin content, while some apple varieties have high anthocyanin content in both the skin and flesh. Effects of red compared with white-fleshed apples were studied in healthy human subjects in a randomized, placebo-controlled, cross-over intervention trial. Twenty-five healthy subjects consumed dried daily portions of the red-fleshed or placebo (white-fleshed) apple for two weeks, followed by one-week washout and further two-week crossover period. During the study, volunteers provided faecal samples for microbiota composition analysis and blood samples for peripheral blood mononuclear cell (PBMC) gene expression analysis. Subtle differences were observed in the faecal microbiota of subjects that were fed the different apples, with significant (*p* < 0.05) reductions in relative abundances of *Streptococcus*, *Ruminococcus*, *Blautia*, and *Roseburia*, and increased relative abundances of *Sutterella*, *Butyricicoccus*, and *Lactobacillus* in subjects after consuming the red apple. Changes in PBMC gene expression showed 18 mRNA transcripts were differentially expressed between the two groups, of which 16 were immunoglobulin related genes. Pathway analysis showed that these genes had roles in pathways such as immunoglobulin production, B cell-mediated immunity, complement activation, and phagocytosis. In conclusion, this study shows that anthocyanin-rich apples may influence immune function compared to control apples, with changes potentially associated with differences in the faecal microbiota.

## 1. Introduction

The phrase “an apple a day keeps the doctor away” is thought to have originated in Wales, with a variant of it appearing in print in 1866 [1]. While it is a saying often used in jest, there is some recent evidence to support the notion of limited benefit to regular apple eaters for avoiding the use of health care services [2]. Certainly, apples are one of the fruits regularly incorporated into a Mediterranean-style diet [3,4,5]. Such a diet contains minimally processed fruits and vegetables, and is thought to be associated with a reduced risk of various chronic diseases, including cardiovascular disease, type 2 diabetes, and cancer. It may also lead to a slowing of disease progression [6]. The various phytochemicals contained within fruits and vegetables, including phenolics, flavonoids, triterpenoids and carotenoids, may play a key role in this reduction of chronic disease risk [7,8]. 

The carbohydrate composition of dietary plant cell walls plays a significant role in the development and metabolism of the complex gut microbiota. Pectin is an essential component of plant cell walls. Chung et al. [9] confirmed that potentially beneficial *Faecalibacterium prausnitzii* strains could utilize apple pectin as a growth source. They also found that a different bacterium, the Firmicutes species *Eubacterium eligens* DSM3376, promotes the production of an anti-inflammatory cytokine, at least in cell-based assays. Other authors have shown potentially beneficial effects of the consumption of whole apples as opposed to a clear apple juice on circulating lipoproteins and blood pressure in a group of 23 healthy volunteers [10]. These endpoints are generally considered as biomarkers of cardiovascular disease risk. Thus, the authors concluded that, while the consumption of whole apples appears to show health benefits, drinking clear apple juice will not have the same effects, supporting the potential health benefits of consuming a whole fruit, possibly due to the fibre content [11]. 

The colour of an apple is also associated with health benefits. Polyphenols are a large and diverse class of compounds, and flavonoids, including the pigmented anthocyanins, are the largest and best-studied group of these [12,13]. These pigments are responsible for the red colouring of both the skin and/or flesh of an apple, and may confer health benefits. Indeed, Tu et al. attributed most of the reduced cancer risks seen in various studies to the flavonoid composition of the apple [13]. Andre et al. considered anti-inflammatory properties of extracts from five apple cultivars, selected mainly for variation in polyphenol composition [14]. They compared their effects in two cell-based assays, targeting critical points in the inflammation pathway, namely nuclear factor kappa B (NF-κB) activation and expression of tumour necrosis factor alpha (TNFα), both implicated in the onset and progression of diseases associated with acute or chronic inflammation such as inflammatory bowel disease (IBD) [15] and neurodegenerative diseases [16]. They found that those with high levels of particular types of polyphenols targeted these two different, but important, parts of the inflammatory pathway in different ways. 

Espley and coworkers showed that feeding mice a red-fleshed (flavonoid enriched) genetically modified variant of ‘Royal Gala’ apples not only reduced inflammatory biomarkers, but also beneficially modulated the colonic microbiota [17]. Polyphenols have poor bioavailability, mostly influenced by their complex chemical structures, which raises a question of their direct impact on health [18]. However, the primary mode through which these complex polyphenols are metabolized to smaller phenolic compounds is through the action of gut microorganisms [18]. Therefore, understanding the relationship between polyphenols and the microbes that metabolize them is essential for understanding their potential health benefits.

The effects of the red-fleshed apple in modulating the colonic microbiota in mice [17] is especially noteworthy. There is increasing evidence that the gut microbiome plays a critical role in susceptibility to various diseases that are important in humans [19]. At early life stages, it can impact on human growth and development, and there is also evidence that it may influence susceptibility to obesity at a later age [20,21,22]. In later life, the gut microbiome can also play a significant role in the development and progression of inflammatory bowel diseases, and indeed faecal transplants have been suggested as a possible therapy for some forms of these diseases [23]. The gut microbiome may be involved in cirrhosis of the liver due to non-alcoholic fatty liver disease [24], and may also play significant roles in both cancer development and response to specific therapies [25]; indeed, it significantly impacts the efficacy of several therapeutics [26].

Having demonstrated the beneficial effects of a red-fleshed apple in mouse studies, it now becomes essential to consider their effects in humans. Since genetically modified foods cannot ethically or legally be used in human studies in New Zealand, an initial step in this process was to select a naturally occurring type I red-fleshed apple [27]. The use of systems biology approaches to understand the effects of these apples in non-diseased subjects could enable a meaningful understanding of their potential for either helping to prevent disease or slow its progression [28,29]. In the current study, we investigated the hypothesis that two weeks’ daily consumption of a red-fleshed apple would result in changes to the composition of the gut microbiota (assessed using microbial DNA extracted from faecal samples), and reduced expression of pro-inflammatory genes in peripheral blood mononuclear cells (PBMCs), compared with a white-fleshed control apple, in healthy adults. 

## 2. Materials and Methods 

### 2.1. Preparation of Apples

The high anthocyanin red-fleshed and placebo apples for this study were harvested in New Zealand in March 2014. The red-fleshed apples (Figure 1A; *Malus* × *domestica*, Plant & Food Research selections A194R31T068, A236R02T092, and A358R02T100) were harvested from three trees. Placebo apples (Figure 1B) were *M. domestica* ‘Lemonade™’ (cultivar ‘PremA153’), which were chosen due to their combination of pale skin and white flesh, and because they were available to harvest at the same time as the red-fleshed apples. All fruit came from the Hawkes Bay Research Centre, Plant & Food Research. After removing the core, the apples were sliced and dipped in a food grade 2% calcium ascorbate solution, then freeze-dried. At this stage, all apples of each variety were mixed prior to packaging to ensure consistency of supply for the study. The dried apple slices were packaged under vacuum into daily portion sizes, each equivalent to one apple. All processing took place at the Food Bowl (Manukau, New Zealand) and was according to New Zealand Food Standards.

### 2.2. Analysis of Apple Polyphenols

Liquid chromatography-mass spectrometry (LC-MS) grade acetonitrile, ethanol, methanol and formic acid were purchased from Thermo Fisher Scientific (Waltham, MA, USA) and ultrapure water was obtained from a Milli-Q Intergral3 system (Millipore, Merck KGaA, Darmstadt, Germany).

Freeze-dried apple fruit were provided for analysis. Each sample was the equivalent of a single apple and four samples were analysed per treatment. The sample (approximately 28 g dry weight) was extracted with 500 mL ethanol/water/formic acid (80:20:1 *v*/*v*/*v*) with incubation for 48 h at 2 °C. Following centrifugation and 4× dilution with methanol, the sample extracts were transferred to LC vials and the polyphenol concentrations measured by LC with high-resolution, accurate-MS (LC-HRAM-MS).

The LC-HRAM-MS system was composed of a Dionex Ultimate^®^ 3000 Rapid Separation LC and a micrOTOF-Q II high-resolution mass spectrometer (Bruker Daltonics, Bremen, Germany) fitted with an electrospray ion source. The LC column was an YMC Triart C18 150 mm × 2.0 mm, 1.9 µm (Kyoto, Japan) and was maintained at 40 °C. The flow was 350 µL min^−1^. The solvents were A = 1% formic acid and B = 100% acetonitrile. The solvent gradient was: Initial composition 90% A 10% B, 0–0.5 min; linear gradient to 65% A 35% B, 0.5–15 min; linear gradient to 50% A 50% B, 15–18 min; linear gradient to 100% B, 18–22 min; composition held at 100% A, 22–27 min; linear gradient to 90% A, 10% B, 22–22.2 min; to return to the initial conditions before another sample injection at 32 min. The injection volume for samples and standards was 1 µL. The micrOTOF-Q II parameters were: temperature 225 °C; drying N_2_ flow 6 L min^−1^; nebulizer N_2_ 1.5 bar, endplate offset 500 V, mass range 100–1500 Da, data were acquired at five scans s^−1^. Positive ion electrospray was used with a capillary voltage of 4000 V. Post-acquisition internal mass calibration used sodium formate clusters with the sodium formate delivered by a syringe pump at the start of each chromatographic analysis. Data were processed using QuantAnalysis (Bruker Daltonics, Bremen, Germany).

Polyphenol concentrations were calculated by comparison to external calibration curves of authentic compounds (Sigma-Aldrich, St. Louis, MO, USA). When an authentic compound was not available, the calibration curve of a similar compound was used to calculate equivalents. Polyphenol content was calculated as mg per 100 g of freeze-dried apple.

### 2.3. Clinical Trial Design 

The primary hypothesis was that two weeks’ consumption of a red-fleshed apple, compared with a white-fleshed control apple, would result in significant changes in the composition of the faecal microbiota in healthy adults. Sample size was determined based on published studies in which dietary sources rich in polyphenols were assessed for impacts on relative levels of gut microbes. A difference of approximately 1% (SD 1%) was seen in the levels of *Bifidobacteria* when comparing the consumption of a wild blueberry drink to a placebo drink [30]. Similarly, in a study investigating red wine polyphenols, differences in *Faecalibacterium prausnitzii* and *Roseburia* of approximately 1.3 (log10 copies per gram of feces) were observed in healthy participants, with (SD approximately 1.2) [31]. Using these numbers as a basis, 20 healthy participants would give 90% power of detecting a difference with *p* < 0.05. As the current study was comparing two different apple varieties, the observed differences were expected to be smaller than these reported values. We therefore recruited 30 participants, which allowed for 5 participants to drop out while still expecting to detect significant differences.

This study was a randomized, placebo-controlled, cross-over intervention trial. Thirty healthy volunteers of normal weight (25 female, 5 male; age 20–61 years; body mass index (BMI) 19.4–30.8 kg/m^2^) were recruited by noticeboard advertisements at the University of Auckland and through advertising in local newspapers. Written informed consent was obtained from all subjects. This study was conducted according to the guidelines laid down in the Declaration of Helsinki and all procedures involving human subjects were approved by the New Zealand Health and Disability Ethics Committees (Reference number: NTY11/11/109/AM08). Exclusion criteria were a history of treatment for cancer in the previous five years (excluding non-melanoma skin cancers); a history of gastrointestinal disorders (ulcerative colitis, Crohn’s Disease or Irritable Bowel Syndrome), a history of diabetes, cardiovascular disease, or liver or renal disorders. Other exclusion criteria were: any change in prescribed medication in the previous three months; antibiotics taken in the month prior to the study commencing; current smoker, or have previously smoked more than 10 pack years; consumption of more than four alcoholic beverages a day, or current use of vitamin or mineral supplements.

Once selected for the study, a Food Variety Questionnaire was completed for two weeks prior to the commencement of the study. Participants were then randomly assigned to either consume the red-fleshed or white-fleshed apple, and were provided with sufficient for the first intervention. One portion (equivalent to one apple) of the dried slices of the assigned apple variety was consumed each day for the following two weeks. On their third visit (after washout) participants were provided with sufficient of the second apple variety for the cross-over intervention.

Participants were asked to refrain from eating any apples apart from those provided, but to otherwise maintain their regular diet and exercise levels throughout the study. Participants were also asked to complete a weekly questionnaire on the consumption of the following polyphenol-rich foods: fresh apple, red grapes/red grape juice, cranberries/cranberry juice, berries (including blueberries, raspberries, and blackberries), tea (green or black), red wine, red cabbage, eggplant (with skin on), red-fleshed peaches, red-fleshed plums, or black rice.

For each study visit, following an overnight fast, subjects either attended the Faculty of Medical and Health Sciences, The University of Auckland, or Plant & Food Research, Mt Albert. Visits were on four separate occasions: before the intervention (baseline, week 0); at the end of the first intervention (week 2); after washout and before the second intervention (week 3); and at the end of the second intervention (end of trial, week 5). At each visit, height, weight and waist measurements were taken, and blood and urine samples collected. Participants also provided a faecal sample that had been collected during the 24 h prior to each visit. 

### 2.4. Blood Processing

Venous blood samples were collected into ethylenediaminetetraacetic acid (EDTA)-containing tubes (BD Vacutainer^®^, Becton, Dickinson and Co., Franklin Lakes, NJ, USA), and centrifuged at 3000× *g* for 15 min at 4 °C. Plasma was transferred into microtubes and stored at −20 °C until analysis. Cholesterol (total, high-density lipoprotein (HDL), total/HDL ratio, low-density lipoprotein (LDL) cholesterol), triglycerides, c-reactive protein (CRP) and a full blood count were measured (LabPLUS, Auckland, New Zealand).

### 2.5. PBMC mRNA Sequencing and Analysis

PBMCs were separated from whole blood by centrifugation in a Ficoll-Paque™ density gradient. Cell count and cell viability were assessed using an automated cell counter and trypan blue exclusion staining, respectively. RNA was extracted from PBMCs using an AllPrep DNA/RNA Kit (Qiagen, Germantown, MD, USA) as specified in the manufacturer’s protocol. Following extraction, RNA quality was confirmed using a Bioanalyzer (Agilent Technologies, Palo Alto, CA, USA) and the RNA concentration measured using a nanodrop (NanoDrop Technologies, Wilmington, DE, USA). Only total RNA with a 260:280 nm ratio ≥ 2.0, a 28S:18S peak ratio ≥ 0.8 and an RNA integrity number ≥ 8.0 was used for subsequent sequence analysis. PBMC gene expression was determined by RNAseq using Illumina HiSeq 2 × 125 PE sequencing.

Sequence reads were trimmed using flexbar version 2.4 [32] and mapped against the index prepared from the Human GRCh38 genome using RNA-star version 2.5.0c [33]. The non-directional counts of uniquely mapped read pairs were summed for each gene and analysed using the EdgeR package version 3.10.5 [34] in the R statistical software environment version 3.2.1. Quasi-likelihood negative binomial generalized linear models were generated from the counts within sample type. Pathway enrichment analysis was performed using the Clue Gene Ontology (ClueGO) package for Cytoscape.

### 2.6. Analysis of Faecal Microbial Composition

DNA from 98 faecal samples, from 25 volunteers (4 time-points [apple intervention start and end, with cross-over], two types of apples), were extracted using Nucleospin Soil kits (Macherey-Nagel, Düren, Germany).

Barcode tagged Illumina 16S V3-V4 rRNA gene libraries were prepared and sequenced using the Illumina MiSeq 2X 250 bp PE platform at NZGL (Massey University, Palmerston North, New Zealand). Sequencing data was delivered as demultiplexed fastq files with adapter sequences trimmed. Sequences reads were processed and quality trimmed using QIIME 2 [35,36] with a 25 q-score cut off and paired reads joined using vsearch with a minimum overlap of 20 bp and no mismatched allowed. Remaining reads were denoised and chimera checked using the deblur algorithm. Sequence reads were classified by aligning against the Silva 132 small subunit ribosomal RNA database. Alpha diversity was assessed using the Faith’s Phylogenetic Diversity metric and beta diversity was compared using Principal Coordinate Analysis (PCoA) of weighted unifrac phylogenetic distances. The sampling depth used for alpha and beta diversity analysis was 8000 reads. The total number of paired-end sequences that passed quality filtering was 2,462,081 (max 97,318; min 8028; median 23,469). Partial least squares discriminant analysis (PLS-DA) was performed on community compositions at intervention endpoints (T2 and T4) using the mixOmics package [37] for R.

### 2.7. Statistical Analyses

Differential gene expression in PBMCs was determined using Exact Tests for differences between two groups of Negative-Binomial Counts. Genes were considered differentially expressed with a 1.5-fold or more difference and a *p*-value equal to or less than 0.05.

Differences in relative abundances of faecal microbes were analysed using a linear mixed-effects model with a nested design with patient factor nested with treatment sequence. Taxa with a *p*-value equal to or less than 0.05 were considered significantly different in relative abundance.

Other statistical analyses were performed with SPSS version 25 (SPSS, IBM Corporation, Armonk, NY, USA). Continuous data are presented as mean ± standard deviation (SD).

### 2.8. Sequence Data Access

Sequences reads are available for download from the NCBI Sequence Read Archive, accession number PRJNA716437.

## 3. Results

### 3.1. Apple Polyphenol Content

LC-HRAM-MS analysis showed clear differences in the polyphenol content of the two apple varieties, with catechin, epicatechin, and the procyanidins B1, B2 and B5 being higher in the white-fleshed placebo apple, while cyanidin-3-glucoside, 4-p-coumaryl quinic acid, phloridzin, phloridzin xyloside, quercetin 3-arabinoside and quercetin 3-galactoside were all higher in the red-fleshed apple (all *p* < 0.001; Table 1).

### 3.2. Participant Characteristics

A total of 25 participants completed the study; *n* = 20 female, *n* = 5 male (Figure 2 and Table 2). Baseline characteristics including anthropological measures, and fasting plasma lipids and CRP, are shown in Table 2. There were no significant differences in any of these parameters between participants based on sequence (i.e., between those receiving the red-fleshed apple first compared to those receiving the white-fleshed apple first (Table 2)), nor were there any differences in response to the apple treatment (i.e., an effect of the red-fleshed apple compared to the placebo, Table 3), or after the intervention period as a whole (Table 4).

The data from the completed questionnaires regarding consumption of polyphenol-rich foods do not suggest any difference in the frequency of consumption of these foods between the periods during which the different apples were consumed (i.e., there was no difference in overall consumption of these foods between treatments; data not shown), Therefore, it is unlikely these foods influenced the observed outcomes.

### 3.3. PMBC Gene Expression

Overall, few differences were detected in the PBMC transcriptome profiles of those receiving the different apple interventions; none reached the significance threshold after false discovery rate (FDR) correction. However, when using unadjusted p-values, 18 transcripts were differentially expressed. The use of unadjusted p-values should usually be treated with caution, but in this instance, all but 2 of these transcripts (nuclear receptor subfamily 4 group A member 3 (*NR4A3*), and resistin (*RETN*)) encoded immunoglobulin genes and of these, 14 were from the immunoglobulin variable regions (Table 5). This uniformity in the type of differentially expressed genes provide confidence that differences detected by the transcriptome analysis reflects a true biological difference that can be attributed to the difference apple interventions. Of the 18 genes identified as being differentially expressed (Table 5), only those that are annotated or have known links with each other were included in the network analysis. Therefore, 5 of the identified 18 genes are not shown in Figure 3. Pathway enrichment analysis using ClueGO showed that these genes had roles in pathways such as immunoglobulin production, B cell mediated immunity, complement activation, and phagocytosis (Figure 3).

### 3.4. Faecal Microbiota Composition

Overall, there were few differences in the faecal microbiota composition that could be attributed to the type of apple intervention (Figure 4 and Figure 5). However, some differences in community composition could be discerned by partial least squares discriminant analysis (PLS-DA, Figure 6). Differences observed were subtle, with significant reductions (*p* < 0.05) in relative abundances of genera that included *Streptococcus*, *Ruminococcus*, *Blautia*, and *Roseburia* after consuming the red-fleshed apple (Table 6). In contrast, the relative abundance of other taxa, such as *Sutterella*, *Butyricicoccus*, and *Lactobacillus* was increased in subjects after consumption of the red-fleshed apple (Table 6). 

Faecal samples showed communities typical of human intestinal microbiota (dominated by Firmicutes and Bacteriodetes) that was highly variable between individuals and samples. No significant differences in alpha diversity (Faith’s phylogenetic diversity and Chao1 indices) were observed between interventions (data not shown).

## 4. Discussion

The gut microbiota is known to play an essential role in the transformation of many complex plant polyphenols [38]. There is evidence that the presence of specific microbial species or genera can influence the host’s ability to metabolise particular polyphenols, with the bioactivity and bioavailability of many dietary flavonoids being influenced by gastrointestinal microbiome metabolism [39]. Conversely, although the influence of habitual flavonoid intake on the gut microbiome is poorly understood [39], a range of human, animal and in vitro studies have demonstrated that polyphenols from a range of different sources, including berries [30,40,41], tea [42], red wine [31,42,43] and cocoa [44] can modulate the relative levels of intestinal (generally colonic or faecal) microbes. These changes to the microbiota may include inhibiting potential pathogens and increasing levels of putative beneficial microbes [31].

The different apple interventions in the current study (a high anthocyanin red-fleshed apple and a white-fleshed placebo apple) had only minor effects on the composition of the faecal microbiota, a proxy of the composition of the microbiota in the lower intestine. However, the differences that were observed included genera typically associated with fibre degradation and short-chain fatty acid production, such as *Roseburia* and *Ruminococcus* [45]. These genera were decreased as a result of the red-fleshed apple intervention, which does suggest some potential differences in the fibre content and/or composition between the apple types. *Blautia* was another genus decreased by the red-fleshed apple intervention. *Blautia*, while not typically associated with fibre degradation in humans, has been shown to degrade some digestion resistant carbohydrates in vitro [46], and is responsive to dietary carbohydrate levels [47,48]. However, it has also been shown to be reduced in response to black tea and red wine grape extract in an in vitro gut microbial ecosystem [42], an effect the authors suggested was due to the polyphenol content. This is consistent with the idea that the change in *Blautia* observed in the current study is due to the difference in polyphenol content of the two apples. *Streptococcus*, which was decreased by the red-fleshed apple intervention, has also been shown to have an inverse association with dietary fructose levels [49], which may also point to a differential effect on the gut microbiome according to nutrient composition.

The red-fleshed apple intervention did increase *Lactobacillus* relative abundance. Although the precise mechanism for this increase cannot be established here, numerous studies have highlighted the potential for polyphenols to stimulate expansion or activity of *Lactobacillus* [18,50,51], and our results are consistent with these studies. Lactic acid bacteria such as *Lactobacillus* and *Bifidobacterium* have a long association with health benefits, so the increased relative abundance of *Lactobacillus* after the red-fleshed apple intervention is a notable finding. *Butyricicoccus*, a butyrate producer, was another bacterium associated with health benefits [52,53,54] that was increased in response to consumption of the red-fleshed apple. However, there were other changes to the faecal microbial community after consumption of the red-fleshed apples that could be considered negative or detrimental; *Roseburia*, a prominent butyrate producer [48,55,56] was decreased, and *Sutterella*, a Proteobacteria often associated with disease conditions [57,58], was increased after the red-fleshed apple intervention. These changes reiterate the complexity of the gut microbiome and that changes can rarely be classified as simply “good” or “bad”.

Analysing PBMC gene expression profiles can be useful to identify biomarkers of physiological responses, and these profiles have been shown to respond to diet [59]. In our study, subtle shifts in PBMC gene expression were observed between the different apple interventions. Previous data showed changes in mice, but with a genetically modified organism (GMO) apple. This study has shown similar results, but with a non-GMO apple. These changes were predominantly related to genes encoding the immunoglobulin variable regions. Although these changes were not statistically significant after false discovery rate correction (and should therefore be interpreted with some caution), the common functional motif lends credence to these changes being based on an underlying biological mechanism rather than simply being a type 1 error (i.e., a false positive). Our results suggest that the consumption of polyphenol-rich red-fleshed apples for two weeks can alter immune responses through modulation of immunoglobulin expression. This premise is supported by a previous study in rodents where cocoa polyphenols increased serum immunoglobulin (Ig)G and faecal IgA levels [44]. However, the overall physiological relevance and impact of the observed PBMC gene expression changes remain to be determined.

It is possible that clearer differences may have been observed with a longer duration of intervention. A number of other studies which have investigated the impact of polyphenol consumption have been of a greater duration, however this has not necessarily meant the observation of clearer effects of the intervention. For example, an intervention investigating the impact of a polyphenol-rich tart cherry extract (4 weeks) did not show any impact on the gut microbiome [60]. Conversely, several studies are reported in which significant differences in metabolic and/or microbial responses have been seen with an intervention of two weeks’ duration or less, for example pomegranate juice [61] or cranberries [41]. Thus, we believe that the duration of the current study was sufficient to identify any differences, even though these differences were relatively subtle.

It is possible that the predominance of women taking part in the study and the large age range (20–61years) and BMI (19.4–30.8 kg/m^2^) of the participants may have affected the results of the study. The fact that differences were observed in spite of the age and BMI range mean that these results can be considered as valid to a wide proportion of the general healthy adult population. However, with only five male participants it is not possible to draw any conclusions regarding gender-specific effects, and any future studies would have a more balanced gender ratio to address this.

This study does have some limitations. First, because the two apples used were clearly visually different participants could not be blinded. It is therefore possible that there were effects on the faecal microbiota and/or PBMC gene expression as a result of the participants’ expectations. However, studies in which polyphenols were administered in a blinded manner have shown an impact on the faecal microbiota [62] suggesting it is reasonable to conclude that the results observed here are as a result of the different apple per se. 

With regard to the differences between apple varieties, it is possible that other compositional differences between apples (e.g., fibre) may have had some impact on the observed outcomes, reflected in differences in fibre-degrading species such as *Roseburia*. Although we do not have compositional analyses of the specific apples used in this study, it is known that apple composition varies quantitatively to some extent between different varieties (e.g., [63]). Polyphenols are a class of compound that shows particularly large differences, whereas factors such as fibre content are less variable [64]. Further, the two cultivars in our study varied qualitatively in their anthocyanins (red versus not red) and this is reflected in the large quantitative differences in polyphenol content, and changes in genera such as *Blautia* are consistent with a response to the polyphenols themselves. Thus, we maintain it is reasonable to conclude that at least some of the observed effects were due to the relatively large differences in polyphenol content. 

In addition, the 16S rRNA analyses only provide data regarding microbiota community composition; there is no information relating to the underlying genetic makeup of the microbiome, which would be shown using techniques such as shotgun metagenomics. It is, therefore, possible that differences in the genetic potential of the microbiome occurred. Finally, although they were significant, changes in the composition of the faecal microbiota were relatively modest, and it is not clear whether such changes would have long-term implications for health. Another limitation relates to the use of PBMCs to investigate the effects of foods on the human immune system [65] due to the relative ease of collection. However, because PBMCs represent a heterogeneous mixture of different cell types, including T cells, B cells, natural killer (NK) T cells, and monocytes, impacts on the immune system may not always be clear cut. Circulating PBMCs will also be different from immune cells isolated from the gut or lymphatic tissues.

## 5. Conclusions

Results from this study show that naturally bred, anthocyanin-enriched red-fleshed apples may influence immune function compared to non-enriched apples, which have a different anthocyanin profile, and these changes are potentially associated with differences in the faecal microbiota. Therefore, consumption of these apples may have health benefits, in addition to those associated with the consumption of apples per se. However, the mechanistic links between compositional changes in the faecal microbiota and PBMC immunoglobulin gene expression in this study remain to be determined.

## Figures and Tables

**Figure 1 nutrients-13-01092-f001:**
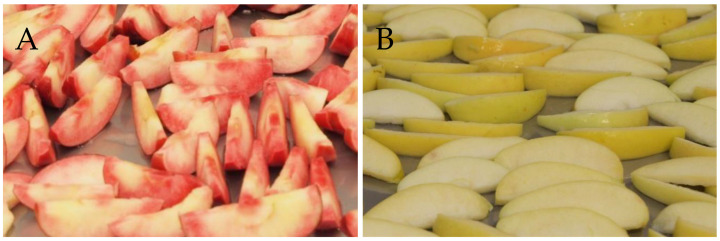
Red-fleshed (**A**) and white-fleshed placebo (**B**) apples after being sliced and dipped in food grade 2% calcium ascorbate.

**Figure 2 nutrients-13-01092-f002:**
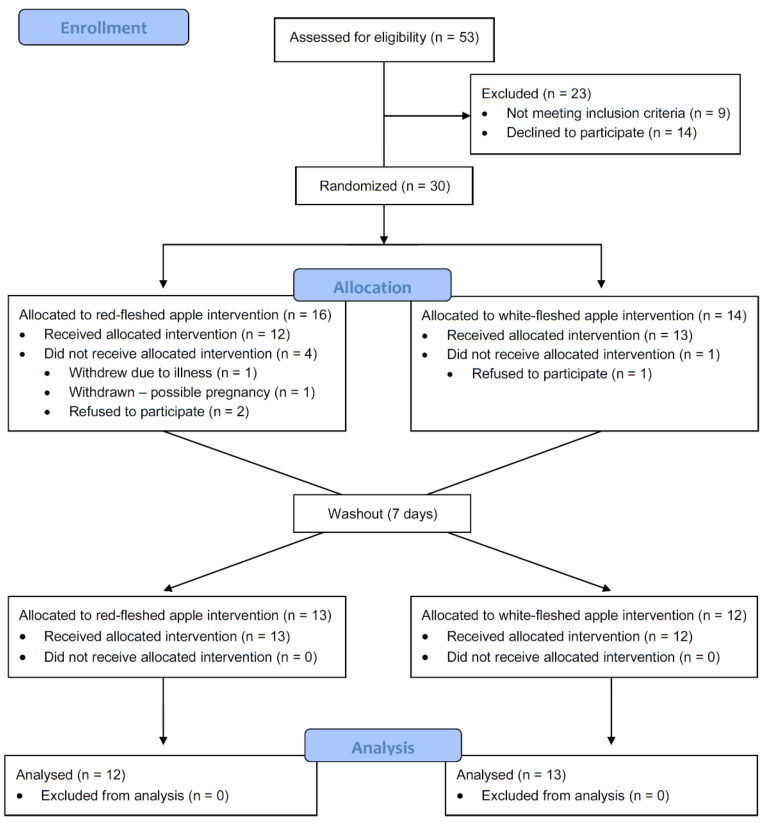
Consort flow diagram of study participant recruitment, intervention, and analysis; n refers to the number of participants at each stage.

**Figure 3 nutrients-13-01092-f003:**
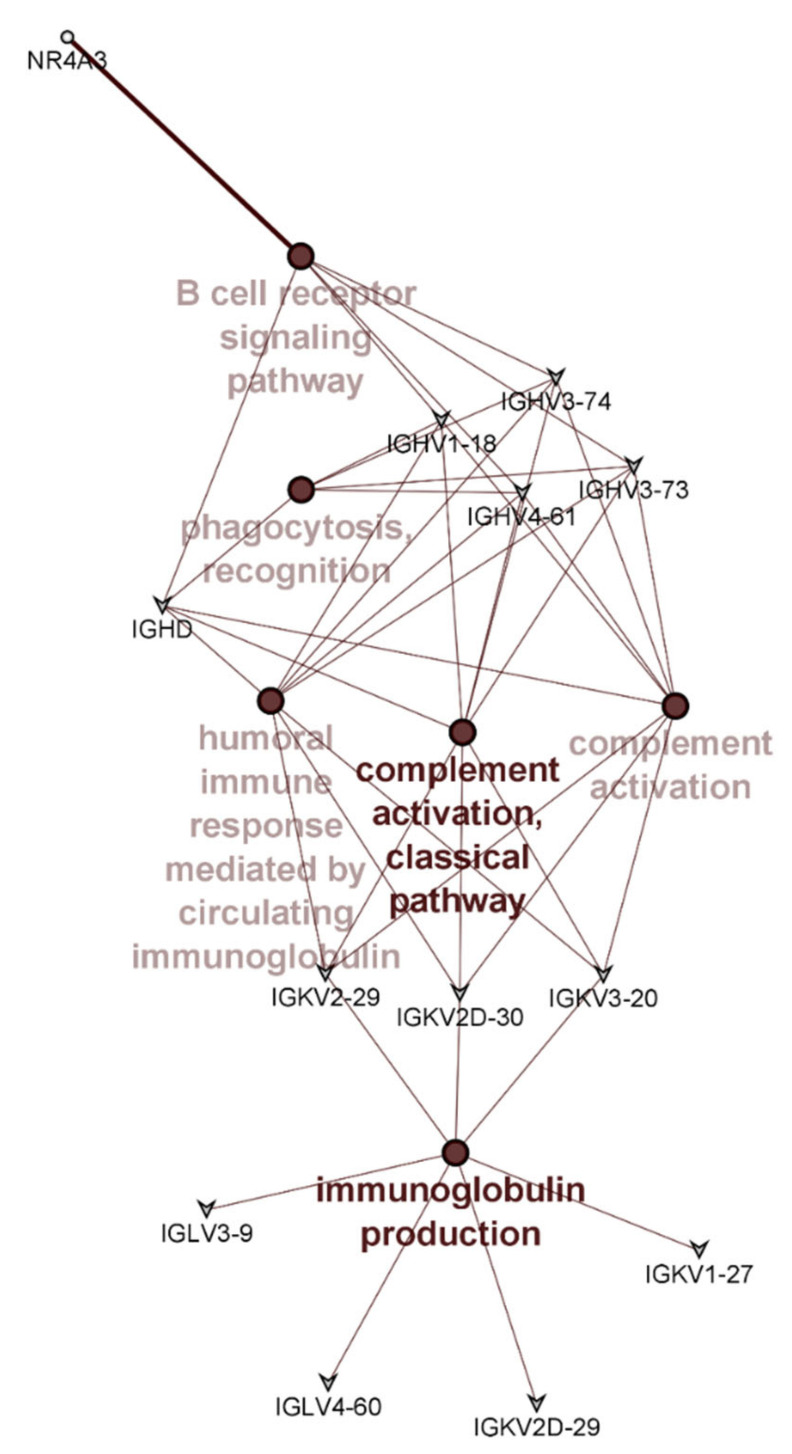
Pathway enrichment analysis showing pathways significantly enriched with differentially expressed genes between the different apple interventions. Colour intensities indicate significance of enrichment with darker colour indicating increasing significance. Genes common to different pathways are shown.

**Figure 4 nutrients-13-01092-f004:**
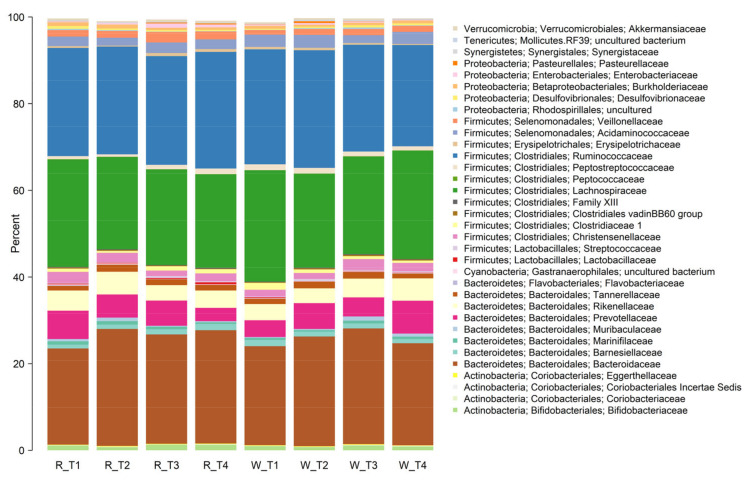
Stacked bar plot showing mean relative abundances of taxa at the family level for those consuming red-fleshed (R) apples and placebo or white-fleshed (W) apples at the start of each intervention period (T1 and T3) and at the end of each intervention period (T2 and T4).

**Figure 5 nutrients-13-01092-f005:**
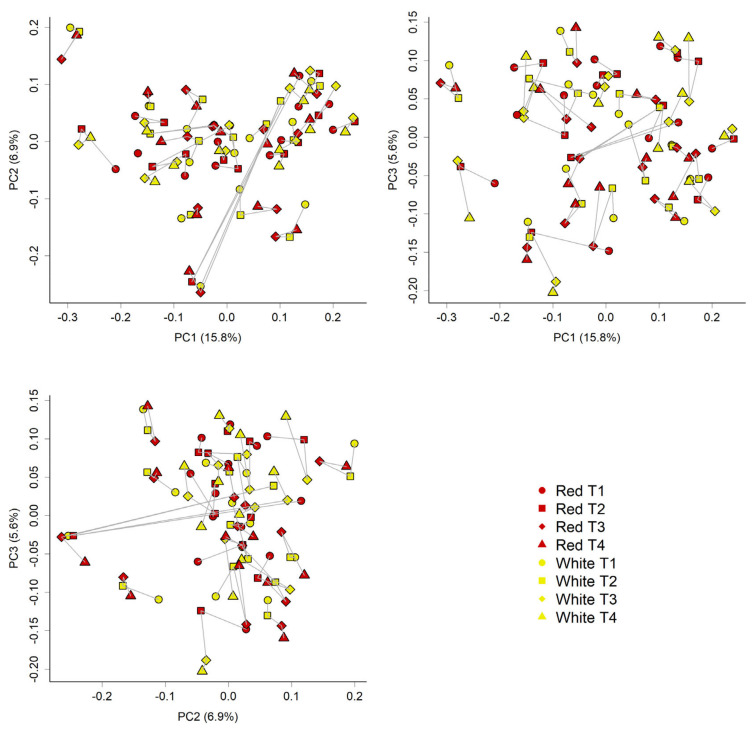
Unweighted unifrac phylogenetic distance analyses of communities reiterate the variability between samples. The Principle coordinate analysis (PCoA) plots below shows the community similarity between samples; Red = red-fleshed apple; yellow = placebo (white-fleshed) apple; spheres = T1 (baseline, week 0); squares = T2 (at the end of the first intervention, week 2); diamonds = T3 (after washout and before the second intervention, week 3); triangles = T4 (at the end of the second intervention which is also the end of trial, week 5). Different plots show PC1 vs. PC2, PC1 vs. PC3, and PC2 vs. PC3. Lines join samples from the same donor. Here, we can see that the donor has the most influence on composition, as would be expected. No obvious differences based on treatments were observed.

**Figure 6 nutrients-13-01092-f006:**
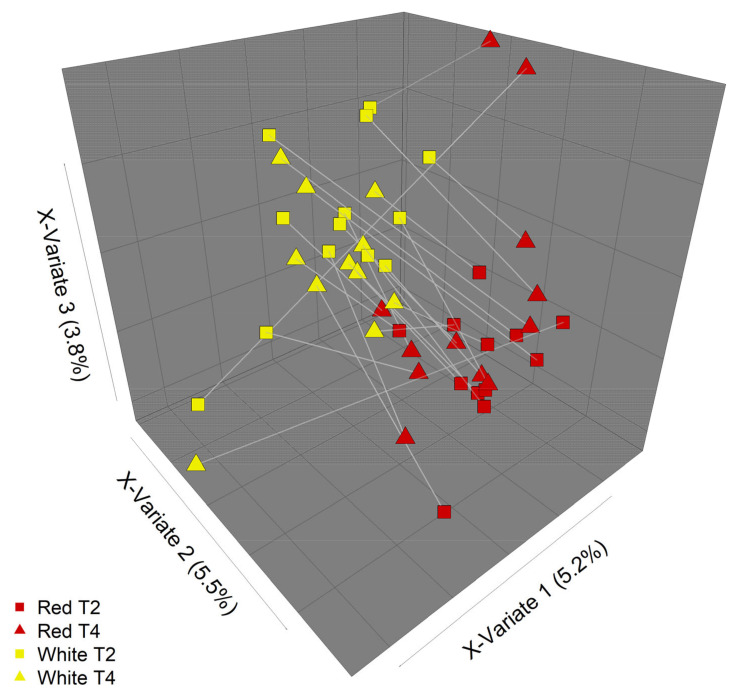
Partial least squares discriminant analysis (PLS-DA) of microbial community composition in response to different apple treatments. Some differences in community composition between the two apple interventions (“Red” = red-fleshed, “White” = white-fleshed) can be discerned by PLS-DA. Plot shows communities at intervention endpoints (T2 and T4). Lines join samples from the same participant.

**Table 1 nutrients-13-01092-t001:** Polyphenol content of the two apple varieties, calculated as mg/100 g freeze-dried apple. Data are expressed as mean ± standard deviation.

Compound	Red-Fleshed	Placebo	*p*-Value
Chlorogenic acid	116.7 ± 13.5	136.9 ± 9.3	0.102
Catechin	0.3 ± 0.0	9.9 ± 1.7	<0.001
Cyanidin 3-galactoside	26.3 ± 3.0	0.0 ± 0.0	<0.001
Cyanidin 3-glucoside	0.3 ± 0.0	0.0 ± 0.0	<0.001
Epicatechin	3.0 ± 0.4	34.7 ± 3.6	<0.001
4-p-Coumaryl quinic acid	0.0 ± 0.0	0.0 ± 0.0	N/A
Phloridzin	11.1 ± 1.1	6.0 ± 1.2	<0.01
Phloridzin xyloside	119.4 ± 10.7	35.2 ± 4.1	<0.001
Procyanidin B1	0.2 ± 0.0	13.2 ± 2.1	<0.001
Procyanidin B2	4.3 ± 0.1	70.0 ± 10.2	<0.001
Procyanidin B5	0.5 ± 0.1	6.6 ± 0.9	<0.001
Quercetin 3-arabinoside	2.1 ± 0.1	1.2 ± 0.3	<0.01
Quercetin 3-galactoside	19.1 ± 0.6	6.5 ± 2.7	<0.001
Quercetin 3-rutinoside	0.9 ± 0.6	0.6 ± 0.3	0.5339

**Table 2 nutrients-13-01092-t002:** Baseline participant characteristics.

Measure	Female (*n* = 20)		Male (*n* = 5)	
Sequence ^1^	1 (*n* = 10)	2 (*n* = 10)	1 (*n* = 2)	2 (*n* = 3)
BMI (kg/m) ^2^	23.3 ± 3.0	23.4 ± 2.1	22.8^2^	27.4 ± 4.7
Age (years)	39.8 ± 15.4	38.0 ± 13.0	47.3 ± 8.1	39.7 ± 14.2
Cholesterol (mmol/L)	4.87 ± 1.18	5.34 ± 1.42	6.25 ± 2.62	5.20 ± 1.65
HDL cholesterol (mmol/L)	1.85 ± 0.36	1.76 ± 0.20	2.25 ± 0.21	1.40 ± 0.46
Chol/HDL ratio	2.66 ± 0.51	3.06 ± 0.94	2.85 ± 1.48	4.03 ± 1.68
LDL cholesterol (mmol/L)	2.65 ± 0.99	3.16 ± 1.36	3.40 ± 2.40	3.17 ± 1.70
Triglycerides (mmol/L)	0.81 ± 0.21	0.94 ± 0.39	1.25 ± 0.92	1.40 ± 0.72
C-reactive protein (CRP; mg/L) ^3^	5.70 ± 12.01	1.05 ± 0.93	1.00 ± 0.00	4.67 ± 7.22

^1^ Sequence refers to the order in which the two apples were consumed; 1 = red-fleshed apple first; 2 = placebo (white-fleshed) apple first. ^2^ As a baseline BMI value was not obtained for one of the male participants, there is no standard deviation (SD) for this value. ^3^ For CRP, several data values were below the lower limit of quantification (LLOQ); for analysis, these were set to ½ LLOQ. Cholesterol, triglyceride and CRP measures are all in plasma. Abbreviations: BMI = body mass index; HDL = high-density lipoprotein; LDL = low-density lipoprotein; *n* = number of participants. All data are presented as the mean ± SD.

**Table 3 nutrients-13-01092-t003:** Effect of red-fleshed apple on fasting plasma cholesterol, triglycerides, and CRP.

Variable	Apple	Estimate (95% CI)	*p*-Value
CRP	Red	−0.372 (−0.976–0.232)	0.2208
	White	0	
Cholesterol	Red	−0.124 (−0.329–0.082)	0.2315
	White	0	
Cholesterol/HDL	Red	0.036 (−0.114–0.186)	0.6337
	White	0	
HDL	Red	−0.096 (−0.203–0.011)	0.0761
	White	0	
LDL	Red	−0.043 (−0.198–0.111)	0.5748
	White	0	
Triglycerides	Red	0.069 (−0.094–0.231)	0.3977
	White	0	

Abbreviations: CI = confidence interval; CRP = C-reactive protein; HDL = high-density lipoprotein; LDL = low-density lipoprotein. All data are presented as the mean ± standard deviation.

**Table 4 nutrients-13-01092-t004:** Effect of the overall intervention on fasting plasma cholesterol, triglycerides, and CRP.

Variable	Intervention	*n*	Mean ± SD	Range	*p*-Value
CRP (mg L^−1^)	Before	48	1.354 ± 1.830	0.5–9.0	0.3648
	After	49	1.286 ± 1.373	0.5–6.0	
Cholesterol	Before	50	5.128 ± 1.359	3.3–8.1	0.0565
	After	49	4.986 ± 1.322	3.1–8.0	
Cholesterol/HDL ratio	Before	50	2.956 ± 0.945	1.7–6.0	0.8714
	After	49	2.953 ± 0.845	1.7–5.6	
HDL	Before	50	1.794 ± 0.363	1.0–2.6	0.1353
	After	49	1.737 ± 0.369	1.0–2.6	
LDL	Before	50	2.914 ± 1.211	1.2–5.7	0.1293
	After	49	2.831 ± 1.144	1.2–5.5	
Triglycerides	Before	50	0.918 ± 0.411	0.4–2.0	0.9594
	After	49	0.918 ± 0.356	0.3–2.1	

Abbreviations: CRP = C-reactive protein; HDL = high-density lipoprotein; LDL = low-density lipoprotein; *n* = number of participants for which data are available. All data are presented as the mean ± standard deviation (SD), with units mmol L^−1^ unless otherwise defined; the range is also shown.

**Table 5 nutrients-13-01092-t005:** Differentially expressed genes between apple interventions.

Ensembl ID	Gene Name	Description	LogFC	LogCPM	*p*-Value
ENSG00000243264	IGKV2D-29	Immunoglobulin kappa variable 2D-29	1.50	2.60	<0.001
ENSG00000244575	IGKV1-27	Immunoglobulin kappa variable 1-27	−1.38	4.00	0.002
ENSG00000211976	IGHV3-73	Immunoglobulin heavy variable 3-73	1.17	2.32	0.002
ENSG00000253998	IGKV2-29	Immunoglobulin kappa variable 2-29	1.05	0.59	0.002
ENSG00000224650	IGHV3-74	Immunoglobulin heavy variable 3-74	0.77	3.73	0.004
ENSG00000119508	NR4A3	Nuclear receptor subfamily 4 group A member 3	−1.15	1.05	0.010
ENSG00000239571	IGKV2D-30	Immunoglobulin kappa variable 2D-30	0.79	0.81	0.013
ENSG00000211898	IGHD	Immunoglobulin heavy constant delta	0.77	7.06	0.014
ENSG00000104918	RETN	Resistin	0.71	2.85	0.015
ENSG00000239951	IGKV3-20	Immunoglobulin kappa variable 3-20	−0.90	6.79	0.019
ENSG00000211970	IGHV4-61	Immunoglobulin heavy variable 4-61	−1.12	3.16	0.025
ENSG00000211639	IGLV4-60	Immunoglobulin lambda variable 4-60	−1.13	1.90	0.025
ENSG00000211945	IGHV1-18	Immunoglobulin heavy variable 1-18	−0.81	3.23	0.033
ENSG00000211670	IGLV3-9	Immunoglobulin lambda variable 3-9	0.71	1.97	0.033
ENSG00000239855	IGKV1-6	Immunoglobulin kappa variable 1-6	0.64	2.66	0.046
ENSG00000211611	IGKV6-21	Immunoglobulin kappa variable 6-21	0.65	0.88	0.047
ENSG00000211668	IGLV2-11	Immunoglobulin lambda variable 2-11	−0.77	4.25	0.047
ENSG00000211895	IGHA1	Immunoglobulin heavy constant alpha 1	−0.89	9.78	0.048

LogFC = log2 fold change, LogCPM = log2 counts per million. Positive LogFC value indicates higher expression in the placebo (white-fleshed) apple group; negative LogFC indicates lower expression in the placebo apple group.

**Table 6 nutrients-13-01092-t006:** Taxa with significantly different mean relative abundances between apple interventions (linear mixed-effects model)**.**

Phylum	Genus	Red	White	*p*-Value
Firmicutes	*Blautia*	2.53 ± 0.32	3.12 ± 0.32	0.049
Firmicutes	*Roseburia*	2.30 ± 0.26	3.07 ± 0.35	0.041
Firmicutes	*Phascolarctobacterium*	1.75 ± 0.33	2.79 ± 0.59	0.045
Firmicutes	*Ruminococcus* 1	1.10 ± 0.21	1.69 ± 0.28	0.024
Proteobacteria	*Sutterella*	0.50 ± 0.12	0.33 ± 0.07	0.026
Firmicutes	*Lactobacillus*	0.30 ± 0.19	0.04 ± 0.02	0.045
Bacteroidetes	*Coprobacter*	0.27 ± 0.18	0.11 ± 0.04	0.001
Firmicutes	*Butyricicoccus*	0.23 ± 0.04	0.18 ± 0.03	0.049
Firmicutes	*Streptococcus*	0.18 ± 0.05	0.36 ± 0.08	0.015
Firmicutes	*Intestinibacter*	0.17 ± 0.04	0.25 ± 0.05	0.050
Proteobacteria	*Haemophilus*	0.12 ± 0.12	0.24 ± 0.17	0.037
Tenericutes	uncl. Mollicutes RF39	0.11 ± 0.03	0.21 ± 0.06	0.033
Firmicutes	*Terrisporobacter*	0.09 ± 0.03	0.22 ± 0.09	0.042
Tenericutes	unclassified Izimaplasmatales	0.03 ± 0.02	0.09 ± 0.06	0.041
Firmicutes	Uncl. *Ruminococcaceae* UCG 011	0.02 ± 0.01	0.01 ± 0.00	0.040

## Data Availability

Sequences reads for 16S rRNA (faecal microbiota) and RNASeq (PBMC gene expression) are available for download from the NCBI Sequence Read Archive, accession number PRJNA716437.

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
