# Peer review of "A Polyphenol Enriched Variety of Apple Alters Circulating Immune Cell Gene Expression and Faecal Microbiota Composition in Healthy Adults: A Randomized Controlled Trial"

_nutrients, 2021, doi:10.3390/nu13041092_

Round 1
Reviewer 1 Report
The review of the manuscript (manuscript ID: nutrients-1155452) is in the file below.

Author Response
We once again thank the reviewer for their time and effort in assessing our manuscript, and for the suggestions for improvement. We respond specifically as follows:
- Lines 13, 17, 119, 123, 209, 354, use of the "&" symbol: In general, we have correct this as suggested by the reviewer. The only exception is with the use of the term "Plant & Food Research" (lines 13, 119, 123, and 209), because this is the official name as used by that institute.
- Lines 116, 132, 223, 241, 272, 276, 284, 312, 335 (capitalization of sub-headings): These have now all be corrected as suggested.
- Lines 180, 217, 240, 299-311, 302-305, 433 (abbreviations): The abbreviations noted by the reviewer have now all been corrected.
- Line 242 (participant number): This has now been corrected.
- Lines 263, 267, 372 (p-value): This has been corrected, and p-value (with lower case p) now used throughout.
- Line 301 (Table 2) - SD value: It has been clarified that, because of the two participants in this group, a BMI value at baseline was not obtained for one, there is no standard deviation value available.
- Line 309 (Table 4): In the header of Table 4, "N" has been changed to "n" which is defined under the table as the number of participants. "P" has been changed to "p-value" as in point 5 above. The "CI" abbreviation has been removed.
- Line 309 (Table 4): Units for the data have now been defined.
- Lines 328, 351, 355, 365 (Figure fonts): We understand that the fonts used in the Figures are acceptable.
- Lines 332-334 (Figure 3): As suggested by the reviewer, we have moved this text from the legend for Figure 3 into the main manuscript results (now lines 326-329).
- Line 402 (in vitro): This is now italicized (line 407 of the revised manuscript).
- Lines 545-726 (References): We have checked the references, particularly with regard to the points raised by the reviewer, and corrected them as necessary. For all manuscript titles, the first letters are now capitalized. With regards to the use of accents (such as the tilde and acute), these are frequently not included in the citation as downloaded. We thank the reviewer for pointing these out, and have ensure they are now included.
Reviewer 2 Report
Indeed the new version of the manuscript of Barnett M et al., has significantly been improved. The manuscript is easier to follow and the results are presented in a more specific way. Overall it is less confusing and the parameters that gave no statistically significant differences are clearly mentioned. The discussion section is more specific and authors have added more relative studies. In my opinion, authors have made the necessary changes and managed to improve the quality of the manuscript.
However, there are some issues relative to the design of the study that authors also ackwoledge and unfortunatelly, cannot be changed. These parameters include sample size, intervention period and the fact that not that many significant results were obtained.
Author Response
We again thank the reviewer for their time in assessing our manuscript, and we appreciate their previous input which has been helpful in improving the manuscript.
We accept that there are issues regarding sample size and duration of the study. While these cannot be changed, we believe we have now appropriately acknowledged and discussed them as limitations within the manuscript.